# Structural basis of substrate recognition and thermal protection by a small heat shock protein

Chuanyang Yu[1,6], Stephen King Pong Leung[1,6], Wenxin Zhang[1], Louis Tung Faat Lai [1], Ying Ki Chan[1], Man Chit Wong[1], Samir Benlekbir[2], Yong Cui[3], Liwen Jiang [1,4,5] & Wilson Chun Yu Lau [1✉]

Small heat shock proteins (sHsps) bind unfolding proteins, thereby playing a pivotal role in the maintenance of proteostasis in virtually all living organisms. Structural elucidation of sHsp-substrate complexes has been hampered by the transient and heterogeneous nature of their interactions, and the precise mechanisms underlying substrate recognition, promiscuity, and chaperone activity of sHsps remain unclear. Here we show the formation of a stable complex between *Arabidopsis thaliana* plastid sHsp, Hsp21, and its natural substrate 1-deoxy-D-xylulose 5-phosphate synthase (DXPS) under heat stress, and report cryo-electron microscopy structures of Hsp21, DXPS and Hsp21-DXPS complex at near-atomic resolution. Monomeric Hsp21 binds across the dimer interface of DXPS and engages in multivalent interactions by recognizing highly dynamic structural elements in DXPS. Hsp21 partly unfolds its central α-crystallin domain to facilitate binding of DXPS, which preserves a native-like structure. This mode of interaction suggests a mechanism of sHsps anti-aggregation activity towards a broad range of substrates.

[1] State Key Laboratory of Agrobiotechnology, School of Life Sciences, The Chinese University of Hong Kong, Shatin, New Territories, Hong Kong, China. [2] The Hospital for Sick Children Research Institute, Toronto, Canada. [3] State Key Laboratory of Cellular Stress Biology, School of Life Sciences, Xiamen University, Xiamen, China. [4] Centre for Cell and Developmental Biology, and Institute of Plant Molecular Biology and Agricultural Biotechnology, The Chinese University of Hong Kong, Shatin, New Territories, Hong Kong, China. [5] CUHK Shenzhen Research Institute, The Chinese University of Hong Kong, Shenzhen, China. [6]These authors contributed equally: Chuanyang Yu, Stephen King Pong Leung. ✉email: wcylau@cuhk.edu.hk

Small heat shock proteins (sHsps) represent a class of highly conserved molecular chaperones that prevent irreversible unfolding or aggregation of proteins under stress conditions to maintain protein homeostasis[1,2]. Known as "housekeeping" proteins, some sHsps can reach up to 1% of total cellular proteins in response to heat shock, highlighting its fundamental role in thermotolerance and cellular stress response[3]. They function mainly to bind to and sequester non-native and denaturing proteins in an ATP-independent manner, thereby preventing their irreversible aggregation[4]. In contrast to other major chaperone families, sHsps do not process the bound substrate proteins directly, but instead coordinate with other cellular machineries such as ATP-dependent Hsp70 and Hsp100/Clp chaperone systems to facilitate the subsequent disaggregation and/or refolding of the substrates[5,6].

The ability to assemble into oligomers of varying subunit stoichiometries represents a unique feature of sHsps[7]. Each sHsp monomer possesses a tripartite domain architecture comprising of a N-terminal region (NTR), a conserved α-crystallin domain (ACD) and a C-terminal region (CTR)[8-10]. The central ACD is composed of anti-parallel β strands, forming a structured, immunoglobin-like β-sandwich domain[11,12]. Both the NTR and CTR exhibit high flexibility, extensive sequence variation and contribute to oligomerization of sHsp[10,13,14]. Notably, sHsp oligomers can undergo rapid subunit exchange with the rate of exchange being temperature-dependent, and this structural plasticity is likely essential for the chaperone activity[15-17].

Activation of sHsp is generally thought to involve the disassembly of inactive oligomers to active sub-oligomeric species through poorly understood mechanisms[14,18-20]. Furthermore, sHsps are promiscuous chaperones capable of recognizing a broad spectrum of non-native substrate proteins[21]. Although studied extensively, detailed understanding of substrate recognition conferred by sHsp remains very limited.

In this work, we study Hsp21, a chloroplast-localized sHsp in all photosynthesizing plants that plays a critical role in chloroplast development, fruit maturation, thermomemory regulation and protection of the photosynthetic complex during heat stress[22-27]. Through identification of a client substrate of Hsp21, 1-deoxy-D-xylulose 5-phosphate synthase (DXPS), we isolate the Hsp21-DXPS complex under heat stress conditions and determine its structure by single-particle cryo-electron microscopy (cryo-EM). The structure reveals how partial unfolding of a sHsp confers thermoprotection to its client substrate.

## Results

### Hsp21 prevents the aggregation of DXPS in vivo and in vitro.

A previous report demonstrated the existence of a chloroplast unfolded protein response in Arabidopsis, and that inhibition of the plastome gene expression triggers a retrograde signaling pathway, resulting in the expression of various other plastid-targeted chaperones including Hsp21 to help restore the proteome balance[28]. In particular, plastome gene expression inhibition has been shown to also promote the accumulation of aggregated DXPS as a result of disruption and consequently reduction of the activity of the Clp proteolytic complex. DXPS catalyzes the first and rate-limiting step of the methylerythritol 4-phosphate (MEP) pathway responsible for the synthesis of metabolic precursors for isoprenoids in plants[29]. Because the MEP pathway is absent in mammals but is present in most pathogenic bacteria, plants, and malaria parasites, DXPS is also an attractive target for the development of antibiotics, herbicides, and antimalarial drugs[30]. Taking advantage of the aggregation properties of DXPS, we sought to test whether Hsp21 promotes DXPS disaggregation in vivo by transiently co-expressing

Hsp21and DXPS fusion proteins in Arabidopsis protoplasts and then analyzing the accumulation of aggregated DXPS proteins upon pharmacological inhibition of plastome gene expression by treating cells with lincomycin, which specifically inhibits chloroplast translation[31]. Under laser scanning confocal microscopy, overexpression of DXPS and Hsp21 both produced fluorescent punctate patterns. The histologic bases of the punctate patterns observed for DXPS and Hsp21 have been previously attributed to the formation of insoluble protein aggregates caused by overexpression[29,32,33] and the general high localization to the nucleoids[24], respectively. Inhibition of plastome gene expression led to an increase in the total DXPS punctate spot area in the treated cells compared to the untreated cells (Supplementary Fig. 1a, right panel, row 1), consistent with our current understanding that these punctate spots represent aggregated DXPS. Importantly, Hsp21 overexpression significantly suppressed DXPS aggregate formation compared to the control (Supplementary Fig. 1a, right panel, row 3 and Supplementary Fig. 1b, c). Note that the apparent lack of colocalization between Hsp21 and DXPS under confocal microscopy was likely due to fluorescence signals of Hsp21-DXPS complexes being diluted out compared to that of the punctate spots (Supplementary Fig. 1a, left and right panels, row 3). We further performed fractionation analysis to assess the distribution of DXPS and Hsp21 to the soluble and insoluble fractions and observed a similar increase in the suppression of DXPS aggregation when Hsp21 is present (Supplementary Fig. 1d–f). Hence, Hsp21 prevents the aggregation of DXPS in vivo.

Hsp21 has been shown to be essential for the control of plant thermotolerance and thermomemory[24,25,27,34]. In heat-stressed leaves, it was estimated that the level of Hsp21 accumulates to approximately 0.05% of total soluble protein in the chloroplast[35]. To investigate a potential involvement of Hsp21 in the protection of DXPS under heat stress, we monitored the levels of DXPS aggregates in cells exposed to 25 °C and 37 °C following lincomycin treatment, in the absence and presence of Hsp21 (Fig. 1a, left and right panels, row 1 and 3). Quantification analysis of the DXPS punctate area from confocal images and fractionation analysis both showed that there was a temperature-dependent increase in the amount of DXPS aggregates, and the presence of Hsp21 was able to reverse the DXPS aggregation at either optimal or elevated temperature (Fig. 1a–f). In addition to binding to unfolding proteins and preventing protein aggregate formation, sHsps were frequently found to co-aggregate with misfolded proteins in vivo[36-40]. Likewise, Hsp21 was detected exclusively in the insoluble fraction at 37 °C, co-fractionating with the aggregated DXPS (Fig. 1d). We speculate that our inability to detect the interaction between Hsp21 and DXPS in the soluble fraction is due to the presence of only a small portion of the soluble complexes and/or active substrate handoff between Hsp21 and downstream ATP-dependent chaperones in the cell. With the goal of obtaining purified protein complexes for subsequent structure determination, we next assessed the solubility of DXPS in Escherichia coli (Fig. 1g, h). Consistent with the results obtained from plant protoplasts, DXPS was found to be highly insoluble at 37 °C, whereas co-expressing Hsp21 and DXPS together rendered DXPS more soluble. Finally, we showed that Hsp21 retained its ability to suppress the aggregation of DXPS in vitro using purified proteins (Fig. 1i, j), thus demonstrating that the increase of solubility of DXPS is due to its direct interaction with Hsp21. At the molar ratio of Hsp21 monomer to DXPS monomer of 12:1, near complete solubilization of DXPS was observed, in contrast to the ~10-20% soluble DXPS observed in vivo in the presence of Hsp21 (Fig. 1e, h, and j). Altogether, these data confirmed that Hsp21 confers thermoprotection to DXPS under heat stress.

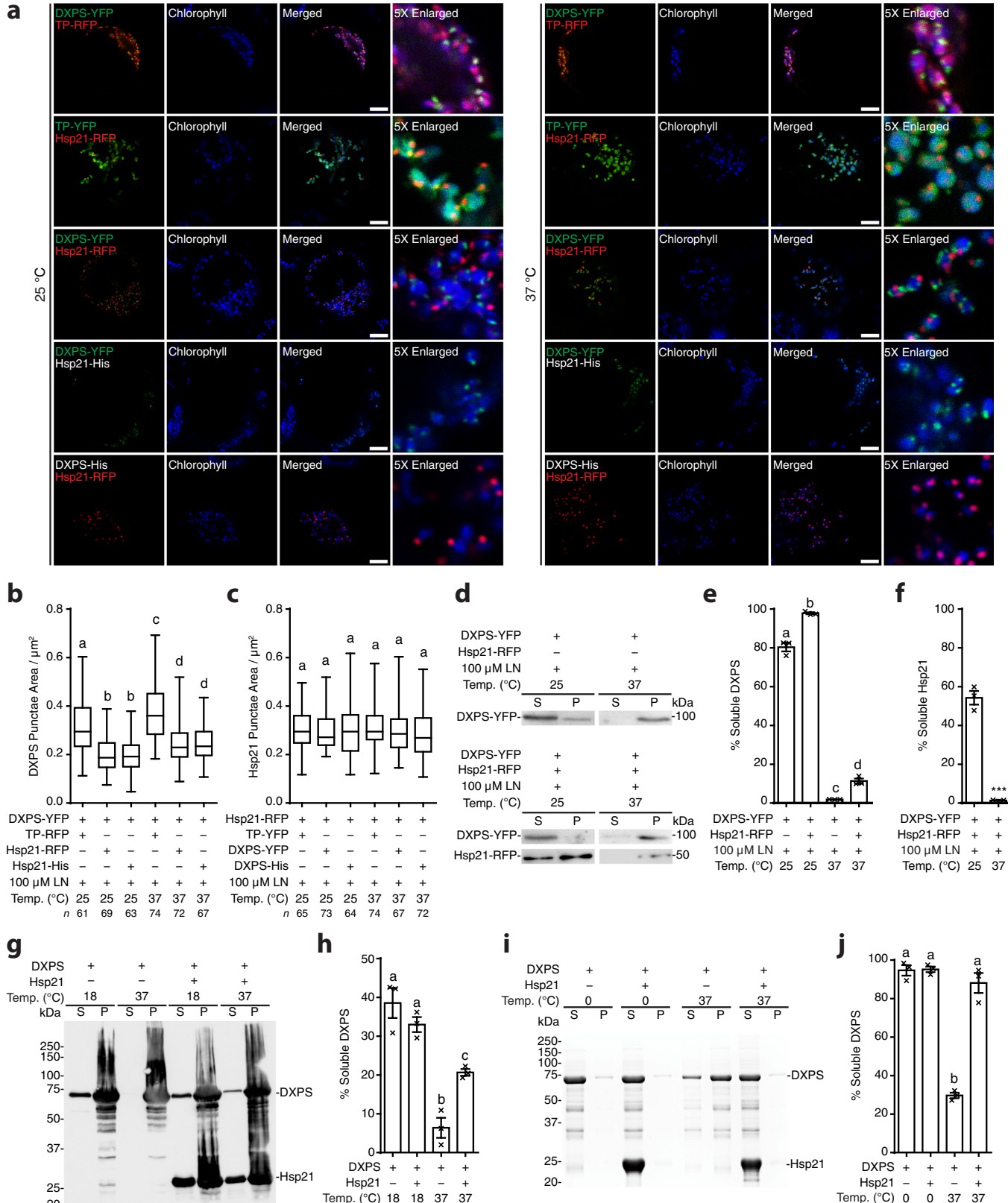

**Cryo-EM analysis of DXPS and the Hsp21-DXPS complex at near-atomic resolution**. Structure characterization of intact sHsp-substrate complexes has proved challenging owing to the transient and heterogeneous nature of their interactions. Here we succeed in obtaining the Hsp21-DXPS complex by co-expression of both Hsp21 and DXPS in *E. coli* at 37 °C under conditions that promote thermally induced aggregation of DXPS and soluble complex formation, and determining its structure by cryo-EM (Supplementary Fig. 2). The amount of DXPS found in soluble complexes isolated by our method represented ~20% of the total expressed DXPS (Fig. 1h). Three-dimensional (3D) classification of the cryo-EM particle images identified subsets of unbound and

**Fig. 1 Hsp21 suppresses DXPS aggregation under heat stress. a** Confocal microscopy images of chloroplasts in *Arabidopsis* protoplasts (treated with 100 μM lincomycin (LN) after transformation and incubated for 2 h at 25 °C or 37 °C before imaging) expressing DXPS-YFP with TP-RFP, Hsp21-RFP or Hsp21-His, and Hsp21-RFP with TP-YFP or DXPS-His. TP: Rubisco activase RecA transit peptide. Scale bar, 10 μm. **b, c** Quantification of (**b**) DXPS punctae area and (**c**) Hsp21 punctae area as shown in (**a**). Values represent mean ± SD, where *n* represented under the graph indicates the number of punctate fluorescent signal from at least two independent experiments. **d** DXPS and Hsp21 protein distribution in soluble and insoluble fractions isolated from *Arabidopsis* protoplasts treated with LN and incubated for 2 h at 25 °C or 37 °C before extraction. **e, f** Densitometry quantification of (**e**) DXPS and (**f**) Hsp21 protein distribution in soluble fractions as shown in (**d**). Values represent mean ± SEM with *n* = 3 biological replicates. **g** DXPS-His-FLAG and His-Hsp21 protein distribution in soluble and insoluble fractions isolated from *E. coli* induced at 18 °C and 37 °C, respectively, detected by Western analysis using an anti-His antibody. **h** Quantification of DXPS-His-FLAG protein distribution in soluble fractions as shown in (**g**). Values represent mean ± SEM with *n* = 3 biological replicates. **i** DXPS-FLAG and Hsp21 protein distribution in soluble and insoluble fractions after mixing purified DXPS-FLAG protein with or without Hsp21 dodecamer protein (Hsp21: DXPS in monomer/monomer molar ratio of 12:1) and incubating for 2.5 h at 0 °C and 37 °C, respectively. Protein distribution was assessed by Coomassie Blue Staining. **j** Densitometry quantification of DXPS protein distribution in soluble fractions as shown in (**i**). Values represent mean ± SEM with *n* = 3 independent experiments. Box and whiskers plots show maxima and minima, upper and lower percentiles (box) and median (line). Letters indicate statistical significance based on one-way ANOVA with post hoc Tukey's multicomparison test ($p \leq 0.05$); means bearing different letters differ significantly. Asterisks indicate statistical significance based on two-tailed unpaired *t*-test (\*\*\**p* = 0.0001). Source data are provided as a Source Data file.

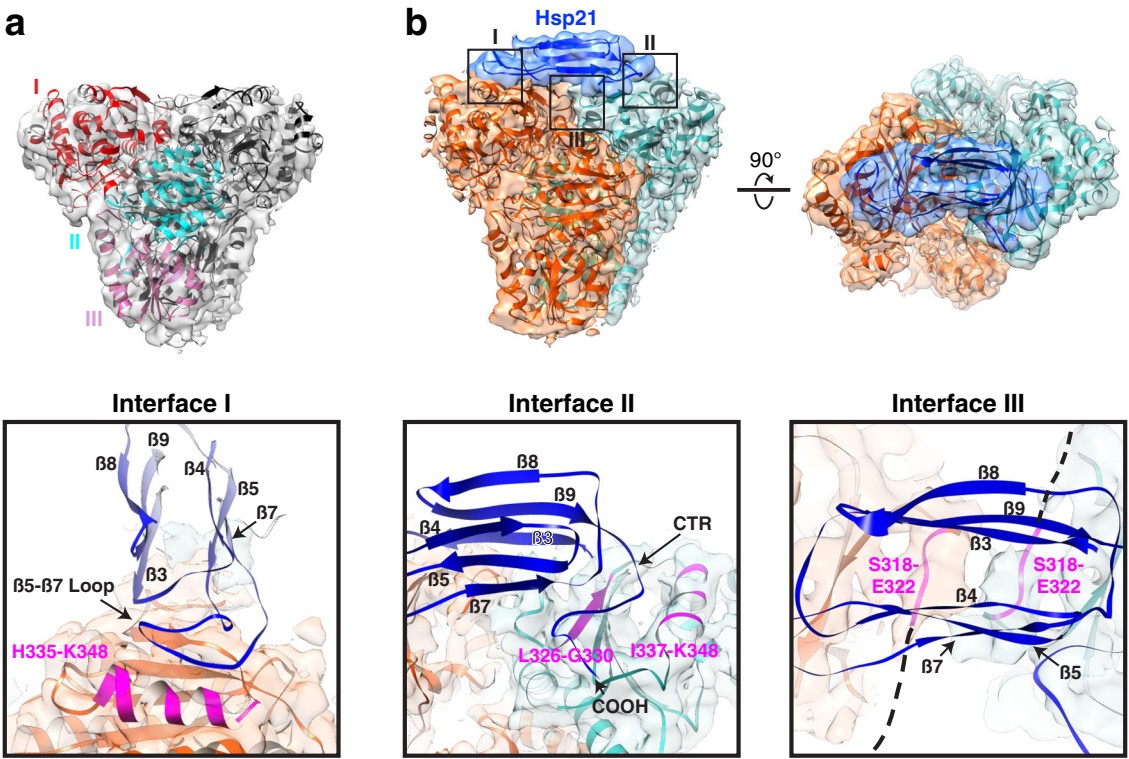

**Fig. 2 Structures of DXPS and Hsp21-DXPS. a** Cryo-EM map of DXPS with the fitted model. The three domains of one subunit of the DXPS dimer are indicated with labels and represented by different color. The other subunit of the dimer is shown in black. **b** Composite cryo-EM map of Hsp21-DXPS with fitted models. The two subunits of the DXPS dimer are shown in orange and sea green and Hsp21 is shown in blue. The three identified interaction interfaces (I, II, and III) are indicated (black boxes) with the corresponding zoomed-in views shown below. Interface III is viewed from the top of the complex. The putative interacting regions on domain I (for Interface I and II) and the mobile loops (for Interface III) of DXPS are highlighted in magenta. The dashed lines denote the disordered mobile loops with no map densities. All β strands and the C-terminus of Hsp21 are labeled. Here DXPS denotes DXPS-His-FLAG and Hsp21 denotes His-Hsp21.

Hsp21-bound DXPS particles (Supplementary Fig. 2e), and the structures of free DXPS and the Hsp21-DXPS complex were determined to a global resolution of 4.0 Å and 3.7 Å, respectively (Supplementary Fig. 2f). The local resolutions vary greatly in both reconstructions, ranging from 3.9 to 5.4 Å and 3.6 to 5.1 Å for DXPS and the Hsp21-DXPS complexes, respectively, likely due to inherent flexibility (Supplementary Fig. 2g). Focused classification and refinement did not improve the local densities, suggesting that the motions were continuous. To facilitate model building, an optimized local sharpening approach was employed to increase the interpretability of the maps[41].

DXPS is a homodimer composed of three domains (Fig. 2a), with domain I being the most dynamic, indicated by the discontinued peripheral densities and the lower local resolution of this region (Supplementary Fig. 2g). Each monomer adopts a distinct conformation, differentiated by the structures of the active sites located at the interfaces between domains I and II of the same monomers (Supplementary Fig. 3a). DXPS catalyzes the oxidative decarboxylation of pyruvate to form 1-deoxy-D-xylulose 5-phosphate that requires opening and closing of its active sites[30]. The density corresponding to the segment spanning residues 247–274, constituting part of the active site, is fully

visible only in one of the monomers (Supplementary Fig. 3b). This segment, previously reported to be flexible[30], likely undergoes disorder-to-order transition and acts as a gate to regulate substrate access to the active site. This observation is in agreement with the enzyme simultaneously populating closed and open conformations in the ligand-free state that were revealed in kinetic studies[30,42]. Moreover, the closed and open conformations resemble those observed in the pre- and post-decarboxylation states of *Deinococcus radiodurans* DXPS, respectively (Supplementary Fig. 3c). Opening of the active site has been postulated to occur along with the displacement of the linker region (spanning residues 288–328 between domains I and II) from the active site. In the structure, the linker regions on both subunits are arranged in a configuration similar to that observed in the closed, pre-decarboxylation state of *Dr*DXPS[43], in spite of the active site conformation (Supplementary Fig. 4). Our data therefore suggested that there is no coordinated movement between active site opening and the repositioning of the linker region. Immediately after the segment 247–274, the density corresponding to a large surface loop of 52 residues long (residues 275–326), referred to as the mobile loop hereafter, is also missing in our structure due to flexibility (Fig. 2a and Supplementary Fig. 5a). Part of the density can only be observed by rendering the map at a very low threshold (Supplementary Fig. 5b). Likewise, the majority of the corresponding mobile loops in the *Dr*DXPS crystal structure are too disordered to yield traceable electron densities[43] and almost certainly occupies multiple positions above domains I of both subunits (Supplementary Fig. 5a).

**Substrate recognition**. Analysis of the Hsp21-DXPS structure revealed how Hsp21 recognizes DXPS and stabilizes it from unfolding. In single-cell organisms and plants, the sHsp dimers formed by the ACDs are widely believed to be the stable building blocks of higher order oligomers and the maintenance of this dimer interface is required for chaperone activity[44,45]. Surprisingly, the Hsp21-DXPS structure enabled unambiguous identification of only one Hsp21 monomer bound to DXPS (Fig. 2b). Hsp21 binds across the dimer interface of DXPS in an asymmetric fashion and is perched atop the domains I of both monomers, forming a crown-shaped structure. The density also allowed modeling in the ACD, arranged in the form of two antiparallel three-stranded β sheets (denoted as β3-β9 following the numbering convention generally adopted for sHsp, with β6 missing in our structure), and the CTR. At least three intermolecular contacts contributing to the Hsp21-DXPS interaction are revealed from the structure. The first interface lies between the β5-β7 loop connecting β5 and β7 strands of the ACD of Hsp21 and residues 335–348 within domain I of one DXPS monomer containing an α-helix. The second interface involves the interaction of the CTR of Hsp21 with a surface patch on domain I near the dimer interface on the opposite DXPS monomer. The final interface is defined by the interaction between the hydrophobic groove of the ACD of Hsp21 and residues 318–322 within the mobile loops of the DXPS. This segment spanning residues 318–322 in the mobile loop are disordered in the native DXPS but becomes ordered upon Hsp21 binding. Although side-chain densities are not clearly resolved at this region of the map, in our model, the hydrophobic residues (Leu[320] or Phe[321]) within this segment are positioned to interact with the hydrophobic groove of the ACD. The identified interactions are supported by the conformational changes occurred in DXPS upon Hsp21 binding (Supplementary Fig. 6). Since the chaperone-binding site formed by these interfaces are normally masked by the mobile loops, it is conceivable that binding of Hsp21 to the apical regions of domains I would require large degree of conformational adjustment of the mobile loops on DXPS, with part of these segments probably becoming more ordered and occupying some of the unassigned densities around the Hsp21-DXPS interface (Supplementary Fig. 7). Consistent with this notion, the segment 247–274 directly connecting to the mobile loops also undergo structural arrangement as a result of Hsp21 binding, thereby converting one active site from a closed to an open conformation (Supplementary Fig. 3d). A common feature of chloroplast sHsps, along with mitochondrial and the dual-targeted chloroplast-mitochondrial sHsps, is their longest NTRs compared to most other cytosolic sHsps[46]. Hsp21 also contains a unique set of methionines with as yet undefined function[47]. While the general role of the NTR has been shown to be involved in substrate binding[20,36,48–53], owing to its inherent flexibility, the entire NTR is not delineated in our structure and it remains unclear whether the NTR is involved in DXPS binding through transient interactions. Overall, the structure shows that Hsp21 sequesters DXPS in its near-native conformation and specifically binds to and stabilizes the dimer interface of DXPS. Thus, maintenance of the quaternary structure of DXPS likely plays a key role in preventing early misfolding of the enzyme, in line with the fact that the first step of inactivation for most multimeric proteins involves subunit dissociation[54]. The structure also provides direct structural evidence for the role of monomeric sHsp in chaperone activity, as previously proposed based on studies of a mammalian sHsp under controlled in vitro conditions[55,56].

**Cryo-EM analysis of the Hsp21 dodecamer**. We also determined the structure of the Hsp21 oligomer in the absence of a substrate. Initial refinement produced a tetrahedral reconstruction at 5 Å resolution with uniformly scattered densities displayed in the inner cavity of the tetrahedron (Supplementary Figs. 8a–d, 9a). Subsequent refinement by masking out these densities improved the overall resolution of the map to 4.6 Å, resolving a total of twelve identical subunits (Fig. 3a, Supplementary Figs. 8, 9a). Only the densities for the ACD and CTR are visible (except for short stretches of NTRs, as described below), suggesting that the inherently flexible NTRs are confined to the inner cavity to some extent and contribute to the scattered densities. The tetrahedral architecture is different from that of the same complex previously determined at a significantly lower resolution[57] but is consistent with the geometry obtained for several sHsp dodecamers[44,58,59]. We also obtained the same structure using an untagged version of Hsp21, arguing against the potential influence of the affinity tag on the oligomer assembly (Supplementary Fig. 10)[60].

Figure 3 shows the molecular details of Hsp21 oligomerization and comparison to the sHsp Hsp16.9 from wheat. Oligomerization of Hsp21 is primarily mediated via the dimeric and non-dimeric interfaces[44]. At the dimeric interface, the β5-β7 loop engages in interaction with β-strands 2 and 3 of the opposite ACD through limited hydrogen bonding (Fig. 3a, middle). The unmodelled terminal residue of the opposite NTR (A127) is also in direct contact with the β5-β7 loop and could further contribute to dimerization. This is in contrast with the reciprocal strand swapping between partner subunits in Hsp16.9 and other previously characterized non-metazoan sHsps[45] and Hsp21 therefore lacks the canonical β6 strand (Fig. 3b). This atypical mode of dimerization presumably leads to a weaker dimeric interface and likely promotes the monomerization of Hsp21 at elevated temperature and during substrate binding. The non-dimeric interface, on the other hand, involves binding of the Val[181] and Ile[183] residues within the conserved I*X*V*X*I oligomerization motif in the CTR to the hydrophobic groove of the ACD formed between β4 and β8 strands of the neighboring monomer

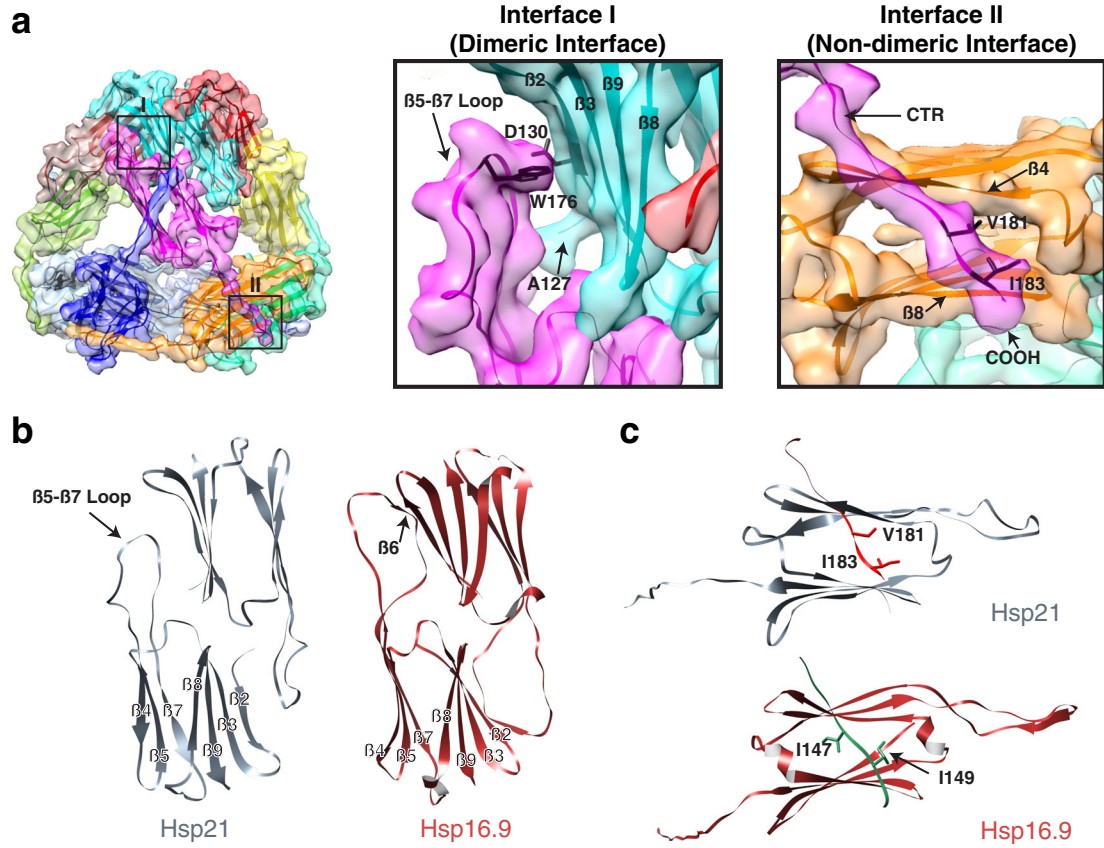

**Fig. 3 Structure of the Hsp21 dodecamer, molecular details of Hsp21 oligomerization and comparison to *Triticum aestivum* Hsp16.9. a** Cryo-EM map of the His-Hsp21 dodecamer with the fitted model. Each subunit is shown in a different color. Interface I (Dimeric interface) and II (Non-dimeric interface) are indicated (black boxes) with the corresponding zoomed-in views shown on the right. The putative interacting residues on opposite subunits (D130 and W176) through limited hydrogen bonding are labeled. The first residue of the ACD (A127) on one subunit presumably also contributing to Hsp21 dimerization is labeled. **b** Comparison of the dimers of Hsp21 (gray) and *T. aestivum* Hsp16.9 (brown) (PDB 1GME). β6 is missing in Hsp21 and Hsp21 does not form reciprocal swapped dimers. Residues 215–227 on Hsp21 and residues 1–42 and 137–151 on Hsp16.9 are deleted for clarity. **c** Comparison of the non-dimeric interfaces of Hsp21 and Hsp16.9. Only residues 218–227 on Hsp21 (red) and residues 143–151 on Hsp16.9 (green) of the neighboring subunit are shown for clarity. For Hsp21, the residues V181 and I183 of the IXVXI (residues 179–183) oligmerization motif are labeled. For Hsp16.9, the residues I147 and I149 on the IXI (residues 147–149) of oligomerization motif are labeled.

(Fig. 3a, right, and c). In the Hsp21-DXPS structure, residues in the segment 318–322 of DXPS bind to the hydrophobic groove on the opposite side of this β-sandwich (Fig. 2b), despite utilizing a similar binding mode.

Within the inner cavity of the dodecamer, extra ordered densities attributable to short stretches of the NTRs are associated with β7 strand of each subunit (Supplementary Fig. 9b, c). Equivalent interactions were seen in the Hsp16.9 crystal structure, where the distal residues from the ordered NTR interact with the hydrophobic residues on β7 strand, either within the same subunit or with the neighboring subunit (Supplementary Fig. 9c)[61]. The latter type of interaction could exist in Hsp21 and additionally stabilize the dodecamer.

**Probing the dodecamer-monomer dissociation of Hsp21.** Using a chemical crosslinking assay, we biochemically probed the otherwise rapid dodecamer-to-monomer transition of Hsp21 with respect to temperature. Our result indicated that while dodecameric Hsp21 was primarily produced by 3,3′-dithiobis (sulfosuccinimidylpropionate) (DTSSP) crosslinking at 25 °C and 37 °C, a significant amount of the monomeric form was only produced at higher temperature (Supplementary Fig. 11a). Furthermore, incubation of Hsp21 with DXPS reduced the amount of the Hsp21 monomers in SDS-PAGE gel and simultaneously

produced the Hsp21-DXPS complex (Supplementary Fig. 11b), which was confirmed by mass spectrometry (Supplementary Table 1). The structural perturbation of the Hsp21 dodecamer leading to an overall size reduction of the oligomer induced by elevated temperature was also directly visualized by negative stain electron microscopy (EM) (Supplementary Fig. 12), providing further consistent evidence for Hsp21 dodecamer dissociation at high temperature. Collectively, these results support our cryo-EM analysis and provide evidence for the heat-induced monomerization of Hsp21.

**Hsp21 undergoes partial unfolding upon binding of DXPS.** The map permitted the construction of a model of the ACD with seven β strands (denoted as β2- β9, with β6 missing in our structure), in striking contrast to the six-stranded ACD structure identified in the Hsp21-DXPS complex, suggesting that the ACD partly unfolds to peel away the first β strand (β2) during its interaction with DXPS (Fig. 4). The loss of this first β strand is expected to destabilize the dimeric interface and likely coupled to the monomerization of Hsp21. Being directly connected to the inherently flexible NTR, the detached β2 strand likely becomes very mobile and is not clearly resolved in the structure, which resonates with the dynamic nature of the β2 strand as observed in several sHsps[48,62–64]. Comparison of the ACDs

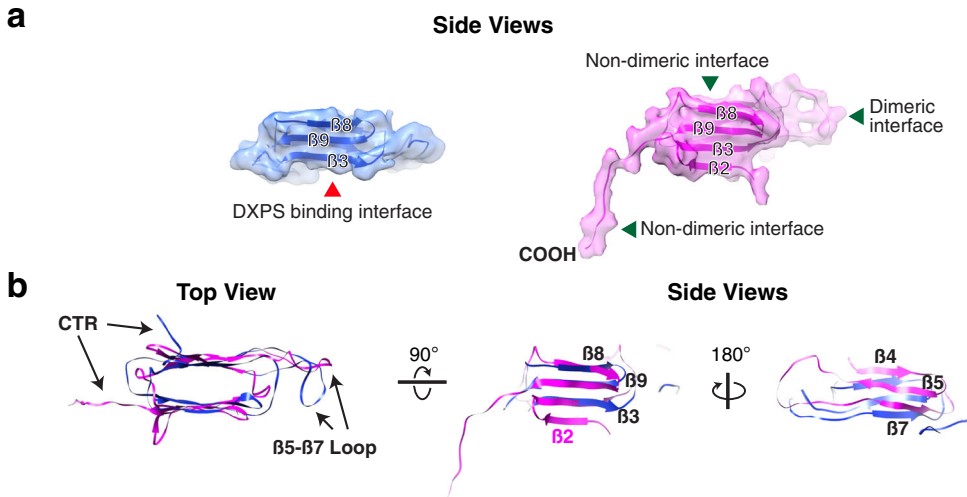

**Fig. 4 Structure comparison between Hsp21 monomers from Hsp21-DXPS and the Hsp21 dodecamer. a** Side view of map segments fitted with the model of the Hsp21 monomer from Hsp21-DXPS (left, blue) and the Hsp21 dodecamer (right, magenta). The beta strands on one sheet of the ACDs are labeled for side-by-side comparison. Note the missing of the beta2 strand in the Hsp21 monomer (blue) derived from the Hsp21-DXPS complex. The DXPS binding, dimeric and non-dimeric interfaces of Hsp21 are labeled. **b** Superimposition of the two models viewed from different directions. For the side views, the back side of ACD was clipped for clarity. The β5-β7 loops and the CTRs on both monomers which they exhibit different conformations are labeled. Twisting of the beta sheets of the ACD from the Hsp21 dodecamer is revealed through overlay of the models (middle and right).

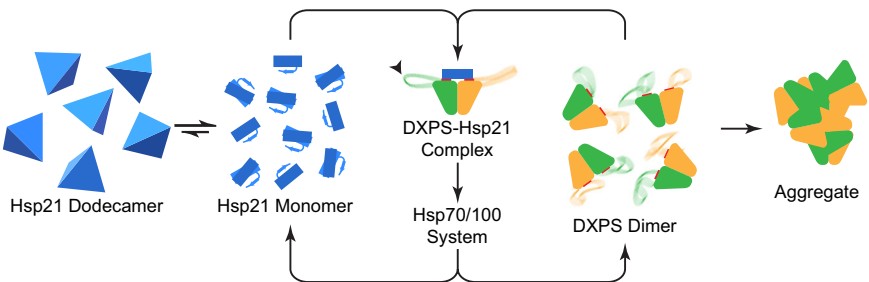

**Fig. 5 A model of the anti-aggregation activity of Hsp21 towards DXPS under heat stress conditions.** The binding surfaces for Hsp21 on DXPS that are normally masked by the mobile loops are indicated with solid red lines. Facilitated by the inherent flexibility of the mobile loops (indicated by gray arrow), the Hsp21 binding surfaces on DXPS (indicated by red lines) are exposed more frequently under elevated temperature. Hsp21 dodecamer dissociation, partial unfolding that results in the loss of β2 strand and the conformational changes occurred upon DXPS binding are shown schematically. The Hsp21-DXPS interaction is predicted to involve both induced fit and conformational selection binding mechanisms.

between the DXPS-bound and DXPS-free states uncovers major conformational transitions in the core β-sandwich domain accompanying substrate binding, from a natural "twisted" state to an "aligned" state where the two β-sheets are aligned near parallel to each other (Fig. 4b, middle and right). This is highly analogous to early events in the unfolding pathway of the fibronectin type III module, a domain that is structurally related to ACD[65,66]. In addition, both the β5-β7 loop and the CTR assume a different conformation from that observed in the substrate-bound form (Fig. 4b, left). Our observation is further supported by molecular dynamics simulations which showed increased propensity for monomeric sHsp to deform and lose the first ß strand when dissociated from the dimer[21]. Our data reveal how DXPS binding elicits conformational changes in different regions of the Hsp21 structure and that structural plasticity is important for its chaperone function.

## Discussion
On the basis of the structural analysis, we propose a mechanism for the anti-aggregation activity of Hsp21 towards DXPS under heat stress (Fig. 5). In response to elevated temperature, DXPS undergoes thermal fluctuations during which the mobile loop samples multiple conformations rapidly, thereby transiently

exposing the chaperone-binding site. Higher temperature also increases the rate of dissociation of the Hsp21 dodecamer via the labile interfaces, followed by its partial unfolding concomitant with a substantial conformational reorganization to accommodate DXPS. Notably, the formation of partially unfolded intermediates at higher physiological temperature has been similarly reported for the isolated fibronectin type III monomer[67]. Hsp21 binding thus involves the selection of the chaperone-binding site on DXPS from ensembles of thermally accessible states as well as structural changes in Hsp21 triggered by multivalent engagement to DXPS. The current model explains the temperature dependence of the Hsp21-DXPS recognition with both conformational selection and induced fit mechanisms playing major roles. Insights obtained from this study provide opportunities for the potential modulation of the MEP pathway through targeting the rate-limiting enzyme and its chaperone. Beyond Hsp21 and DXPS, our study elucidates how a sHsp unfolds to prevent the unfolding of its client substrate and exemplifies a mechanism for fine-tuning the promiscuity of sHsp towards a wide range of client substrates.

## Methods
**Plant growth conditions.** Transient expression of XFP fusion proteins using protoplasts derived from *Arabidopsis* suspension culture cell line PSB-L was

performed as described in our established protocol[68]. Briefly, PSB-L cells were grown in MS medium supplemented with or without 100 μM lincomycin (LN) for 5 days prior to DNA transformation. Transformed protoplasts for confocal analysis were imaged after 2 days of incubation with or without 100 μM LN, respectively.

**Laser scanning confocal microscopy analysis**. All images were captured using a Leica SP8 confocal microscope equipped with a 63x water lens by Leica Application Suite X[69]. Fluorescent signals were detected with the following excitation and emission wavelengths: YFP (488 nm/495–545 nm), RFP (552 nm/590–650 nm), and chlorophyll autofluorescence (552 nm/650–700 nm). Sequential scanning was used to avoid possible crosstalk between fluorescence channels. Quantification of punctate area was done by using the Measurement tool in ImageJ[70].

**Fractionation analysis of *A. thaliana* protoplast lysate**. Transformed protoplasts with indicated treatments were lysed by resuspending in lysis buffer (50 mM Tris-HCl, pH 8.0, 150 mM NaCl, 2% (v/v) Triton X-100, supplemented with protease inhibitor cocktail (cOmplete EDTA-free; Roche)) and passing through a 1-mL syringe with needle. The cell extracts were then centrifuged at 120,000g for 30 min at 4 °C. The supernatant and pellet were collected for immunoblotting analysis. Densitometry of the band intensities was performed by ImageJ[70].

**Fractionation analysis of *E. coli* lysate**. The *E. coli* transformants were grown at 37 °C and induced by 0.5 mM isopropyl-D-1-thiogalactopyranoside (IPTG) at an OD_{600} of 0.6–0.8. After induction at 37 °C for 2 h or at 18 °C overnight, cultures were harvested and suspended in buffer containing 50 mM Tris-HCl, pH 8.0, 300 mM NaCl. Cells were lysed by sonication, and the supernatant (soluble fraction) and pellet (insoluble fraction) were separated by centrifugation at 17,000g at 4 °C for 20 min. Densitometry of the band intensities was performed by ImageJ[70].

**Expression and purification of the Hsp21–DXPS complex**. The coding sequences for *A. thaliana* Hsp21 (UniProt code P31170) (44-227aa) and *A. thaliana* DXPS (UniProt code Q38854) (59-717aa) without their transit peptides were PCR-amplified from cDNA and cloned into a pETDuet-1 vector to generate versions of Hsp21 and DXPS with an N-terminal hexahistidine tag and a C-terminal hex-ahistidine-FLAG tandem affinity tag, respectively. Primers used for cloning were listed in Supplementary Table 2. The construct was transformed into *Escherichia coli* BL21 Rosetta 2 (DE3) (Novagen) for co-expression. Protein expression was induced by the addition of 1 mM IPTG at an OD_{600} of 0.6–0.8, and the cells were grown for 2 h at 37 °C. Cells were pelleted by centrifugation at 7000g for 10 min, resuspended in lysis buffer (50 mM Tris-HCl, pH 8.0, 300 mM NaCl, and 10% (v/v) glycerol) supplemented with protease inhibitor cocktail (cOmplete EDTA-free; Roche) and lysed by sonication. Cell lysate was clarified by centrifugation at 20,000g for 1 h and passed over a HisTrap HP column (GE Healthcare). The column was washed with 20 column volumes (CV) of lysis buffer containing 50 mM imidazole and then subsequently eluted with lysis buffer containing 500 mM imidazole. The eluate was dialyzed in dialysis buffer (50 mM Tris-HCl, pH 8.0 and 300 mM NaCl) and further purified by Anti-FLAG M2 agarose resin (Sigma). After extensive washing with lysis buffer to removed unbound proteins, pure Hsp21-DXPS was eluted with lysis buffer supplemented with 0.2 mg/mL 3x FLAG peptide. Fractions containing pure proteins were pooled and buffer exchanged using a 50 kDa cut-off Centrifugal Filter Unit (Amicon). Proteins were concentrated, flash-frozen in liquid nitrogen and stored at −80 °C for cryo-EM analysis.

**Expression and purification of the Hsp21 dodecamer**. *E. coli* BL21 Rosetta (DE3) was transformed with a pETDuet-1 plasmid expressing an N-terminally His-tagged Hsp21 and protein expression was induced by addition of 1 mM IPTG at an OD_{600} of 0.6–0.7, and the cells were grown for 4-5 h at 37 °C. Cells were lysed in lysis buffer supplemented with 1 mM PMSF and purified via a Ni-NTA column by following the protocol as described above. Proteins were dialyzed in 50 mM Tris-HCl, pH 8.0 and further purified by a Resource Q column (GE Healthcare) using a linear gradient from 0 to 500 mM NaCl. Untagged Hsp21 was purified by HiTrap Q HP column (GE Healthcare) followed by gel filtration in lysis buffer. Purified proteins were pooled, aliquoted and flash-frozen in liquid nitrogen for storage at −80 °C until use.

**Negative EM analysis**. Protein samples were diluted to a final concentration of 0.03 mg/mL in buffer containing 50 mM Tris-HCl, pH 8.0, 300 mM NaCl. Subsequently, 4 μl of sample was applied to the glow-discharged continuous carbon grids. After washing with ultrapure water for 2 times, the grids were stained with 2% (w/v) uranyl acetate for 2 times. The grids were air-dried and stored in an ambient environment until EM analysis.

**Cryo-EM specimen preparation and data acquisition**. Three microliters of proteins (Hsp21-DXPS at 0.7 mg/mL and Hsp21 at 1 mg/mL) were applied to Quantifoil carbon grids (R1.2/1.3, 300 mesh on copper) previously glow-discharged in air for 20 s. Grids were blotted for 1 s at 4 °C and 100 % relative humidity before plunge-freezing in liquid ethane using a Vitrobot Mark IV (ThermoFisher). Images

were acquired with a Titan Krios G3 electron microscope (ThermoFisher), operated at 300 kV and equipped with Falcon 3EC/4 direct electron detectors. Untilted data was recorded automatically with *EPU* software in electron counting mode with a calibrated pixel size of 1.03 Å (Falcon 4) and 1.06 Å (Falcon 3) for Hsp21-DXPS and the Hsp21 dodecamer, respectively. The exposure rates were 5 electrons per pixel per second with a total exposure of 9.6 s for Hsp21-DXPS on Falcon 4 and 0.8 electrons per pixel per second with a total exposure of 60 s for Hsp21 dodecamer on Falcon 3, both fractionated in 30 frames. The total electron exposure was ~45 e−/Å$^2$ on the specimen with a defocus range of −1.5 to −2.5 μm. Tilted data collection was performed to address the partial preferred particle orientation of Hsp21-DXPS in ice identified during initial image processing. Tilted data was recorded similarly except with the stage tilted at −30° or −45°. A total of 4760 movies (3443 untilted and 1317 tilted) for Hsp21-DXPS and 3344 movies for the Hsp21 dodecamer were collected.

**Image processing**. Movie frames were aligned using MotionCor2 and global and per-particle contrast transfer function (CTF) parameters were calculated using gCTF[71]. CTF parameters for the tilted data were estimated as previously described[72]. Motion-corrected micrographs showing poor signal or significant drift were discarded. All subsequent image processing was performed in RELION 3.0[73]. For the Hsp21-DXPS dataset, a total of 3762532 particles and 915665 particles from the untilted and tilted datasets were autopicked using reference-free two-dimensional (2D) class averages as templates generated within RELION.

The two-dimensional classification was performed separately on the untilted and tilted datasets to remove obvious junks. Using the combined datasets, ab initio model generation and iterative 3D classification identified two classes corresponding to DXPS and the Hsp21-DXPS complex. For each class, bad particles were further removed by iterative 3D classification with the inclusion of several junk classes along with the *bona fide* reference, followed by autorefinement to produce maps with overall resolutions of 4.0 Å and 3.7 Å for DXPS and the Hsp21-DXPS complex, respectively.

For the Hsp21 dataset, a total of 1093343 particles were autopicked. Several rounds of reference-free 2D classification followed by another round of reference-free three-dimensional (3D) classification were carried out to select the single best class consisting of 104126 particles. Since the map emerged from the 3D classification clearly indicates that the complex is a tetrahedron, the subsequent autorefinement was performed with tetrahedral symmetry imposed. The refined map reveals scattered densities that are derived from the flexible NTRs. A final round of focused refinement of the ordered regions consisting of both the ACDs and the CTRs resulted in a map at 4.6 Å resolution.

Because of the small size of the complexes studied, CTF refinement and Bayesian Polishing did not improve the resolution of the reconstructions. All reported resolutions were estimated based on the gold-standard Fourier Shell Correction (FSC) 0.143 criterion taking into account of the convolution effect of the solvent mask between the two half maps[74]. Local map resolutions were estimated using RELION. Figures were generated using UCSF Chimera[75] or UCSF ChimeraX[76].

**Model building**. For DXPS and the Hsp21-DXPS complex, the maps were initially sharpened and locally filtered using the post-processing routine in RELION. The homology model of *Arabidopsis* DXPS was generated by SWISS-MODEL server[77] using coordinates of *Deinococcus radiodurans* DXPS (PDB 6OUV) and fitted as a rigid body into the maps. Both maps exhibit significant local resolution variation. To improve the interpretability of the maps for the subsequent model building, a local sharpening and filtering procedure was carried out through amplitude scaling with reference to the fitted atomic model using LOCSCALE[41]. After model-based local density sharpening, models were further adjusted by molecular dynamics-based flexible fitting using CryoFit in PHENIX[78]. For the Hsp21-DXPS complex, a model of Hsp21 monomer that was generated from the Hsp21 dodecamer map (see below) was fitted as a rigid body in the map using Chimera, followed by manual adjustment and model refinement using ProSMART external restraints and Geman-McClure weighting in Coot 0.9 (pre-release)[79]. The final model of the complex was further refined using iterative rounds of real-space refinement in PHENIX[80] with secondary structure and Ramachandran restraints and model building in Coot. While the EM densities for the core region of DXPS were of excellent quality, the peripheral regions including the bound Hsp21 and domains I of DXPS mostly lacked side-chain densities and the models were therefore truncated as polyalanine.

For the Hsp21 dodecamer, a homology model for Hsp21 was generated by the Phyre2 server[81] using coordinates of *Triticum aestivum* Hsp16.9 (PDB 1GME). The map was sharpened and locally filtered using the post-processing routine in RELION and segmented in Chimera. The model was rigidly fitted into the segment and then further adjusted by using normal mode-based flexible fitting in iMODFIT[82] and manual adjustment in Coot. The final dodecamer model was obtained by rigid body fitting, followed by real-space refinement in PHENIX with secondary structure, Ramachandran and rotamer restraints and model building in Coot. Models were evaluated with Molprobity[83]. The refinement statistics are shown in Supplementary Table 3.

**Expression and purification of DXPS**. *A. thaliana* DXPS (UniProt code Q38854) (59-717aa) were cloned into pETDuet-1 vector to generate a C-terminal FLAG tag on DXPS. DXPS were overexpressed in *E. coli* BL21(DE3) Rosetta 2 and cells were grown at 37 °C until $OD_{600}$ of 0.6–0.8 was reached. Proteins were induced by addition of 1 mM IPTG and the cells were grown overnight at 18 °C. Cells were harvested at 7000$g$ for 10 min, resuspended with lysis buffer (50 mM Tris-HCl, pH 8.0, 300 mM NaCl, and 10% (v/v) glycerol) supplemented with protease inhibitor (cOmplete EDTA-free; Roche), and sonicated. The cell lysate was clarified by centrifugation at 20,000$g$ for 1 h at 4 °C and applied to Anti-FLAG M2 agarose resin (Sigma). The column was then washed with 5 CV buffer containing 50 mM Tris-HCl, pH 8.0, 300 mM NaCl, 5 mM ATP, 20 mM $MgCl_2$ and 10% (v/v) glycerol followed by 5 CV buffer washing containing 50 mM Tris-HCl, pH 8.0, 500 mM NaCl and 10% (v/v) glycerol and 15 CV lysis buffer washing. Protein was eluted with lysis buffer supplemented with 0.1 mg/mL 3x FLAG peptides. Pure fractions were pooled, concentrated by 50 kDa cut-off Centrifugal Filter Unit (Amicon), flash frozen in liquid nitrogen and stored at −80 °C until use.

**Aggregation protection assay in vitro**. Purified DXPS was incubated without or with purified Hsp21 in molar ratio of 1:12 for 2.5 h at 0 °C or 37 °C in a total volume of 30 μL. To separate soluble fractions of proteins after treatment, the sample was centrifuged for 10 min at 20,000$g$ and the supernatant was collected for SDS-PAGE. The pellet fractions were further washed once with 50 mM HEPES, pH 8.0, 300 mM NaCl, centrifuged for 10 min at 20,000$g$ at 4 °C and then analyzed by SDS-PAGE.

**Chemical crosslinking**. DTSSP (Thermo Fisher) was dissolved in crosslinking buffer (50 mM HEPES, pH 8.0, 5 mM $MgCl_2$) immediately before used. Proteins were incubated for 2.5 h at 25 °C or 37 °C before addition of DTSSP (final concentration 3 mM). After 20 min, the crosslinking reaction was quenched by addition of a final concentration of 50 mM Tris-HCl and the crosslinked products were analyzed by SDS-PAGE.

**Mass spectrometry**. Gel was stained with Imperial Protein Stain (Thermo Fisher Scientific) after SDS-PAGE separation. Protein bands were excised for in-gel trypsin digestion (40 ng/μL). Peptide extraction was performed twice with 50% (v/v) acetonitrile/0.1% (v/v) trifluoroacetic acid solution. Mass spectrometry analysis was performed by using Bruker Ultrflextreme MALDI-TOF/TOF spectrometer (Bruker Daltonik Gmbh, Bremen, Germany). For spectra acquisition, 0.5 μL samples were spotted onto a MTP AnchorChip 384BC plate, followed by 0.5 μl matrix solution (1 mg/mL α-cyano-4-hydroxycinnamic acid in 95% (v/v) acetonitrile and 0.1% (v/v) trifluoroacetic acid). Data collection was piloted by Flex-Control software using automatic run. The acquired MS data were searched against The Universal Protein Resource database with species restriction to *Arabidopsis thaliana* by Mascot search engine (Matrix Sciences). Only significant hits defined by Mascot probability analysis and with at least one matched peptide were accepted. The searches were carried out using a peptide mass tolerance of 50 ppm, MS/MS ion mass tolerance of 0.5 Da, one missed cleavage and oxidation of methionine.

**Reporting summary**. Further information on research design is available in the Nature Research Reporting Summary linked to this article.

## Data availability

Cryo-EM maps are deposited in the Electron Microscopy Data Bank under accession numbers EMD-30261, EMD-30262, EMD-30263. Atomic models are deposited in the Protein Data Bank under accession numbers 7BZW, 7BZX, 7BZY. All other relevant data are available upon request from the corresponding authors. Source data are provided with this paper.

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

## Acknowledgements

The authors thank Prof. John Rubinstein for critical reading of the manuscript and providing access to the Titan Krios. We thank Prof. Kam Bo Wong for advice on model building. Cryo-EM data were acquired at the Toronto High-Resolution High-Throughput cryo-EM facility, supported by the Canada Foundation for Innovation and Ontario Research Fund. This work was supported by grants from the Research Grants Council of Hong Kong (14105517 to W.C.Y.L.) and (R4005-18, C4033-19E, C4012-16E, C4002-17G, C4002-20W and AoE/M-05/12 to L.J.), National Natural Science Foundation of China (31670179 and 91854201 to L.J.), CUHK Faculty Strategic Development funding (to L.J.), CUHK Seed Fund Research Support (to W.C.Y.L.) and Research Committee of CUHK Direct Grant for Research (4053182 to W.C.Y.L.)

## Author contributions

W.C.Y.L. conceived the project and supervised the research. C.Y., W.Z., L.T.F.L., Y.K.C., and M.C.W. performed cloning and protein purification. S.K.P.L. performed confocal microscopy. C.Y. and S.K.P.L performed aggregation suppression assays. C.Y. performed negative staining and cryo-EM grid preparation. S.B. performed cryo-EM data collection. W.C.Y.L performed image processing and model building. C.Y., S.K.P.L., W.Z., L.T.F.L., and W.C.Y.L. prepared the figures. W.C.Y.L. interpreted the data and wrote the manuscript with input from C.Y., S.K.P.L., Y.C., and L.J.

## Competing interests

The authors declare no competing interests.
