## [Peer Review File · Nature Communications]

REVIEWER COMMENTS

Reviewer #1 (Remarks to the Author):

This manuscript describes arguably the first structural analysis of a substrate:small heat-shock protein complex, revealing considerable insight into the mechanism by which these proteins work. This is an important piece of work that should be published in Nat Comms.

My main criticism is the the presentation of the structural figures - in general I found it quite hard to "read" them well - seeing the differences the authors allude to (e.g. unstructured B2 strand upon binding, the difference in dimerisation between HSP21 and HSP16.9 etc) is difficult because labelling is sparse, alignments are confusing (and sometimes seem poor). In general, I would urge the authors to look at the structural figures carefully to make sure that everything they wish to convey can be clearly seen, and whether they have labelled strands and relevant residues clearly.

Other minor comments:

- *The authors say DXPS is important for antimalarials etc - presumably not in arabidopsis - perhaps just make clear this is referring tot he human homologue
- *The difference in soluble v pellet change upon adding HSP21 (Fig., 1d) is very subtle. Are the authors sure that the increase from 87% to 92% are real?
- *Fig 3 would be helped by a comparison of the dimers of HSP21 and HSP16.9
- *The discussion about the B2 strand being disordered upon binding is interesting in light of the findings of Collier et al, Sci Adv, 2019 (eaav8421) which noted phosphorylation dependent disorder in the B2 strand of HSPB1

Reviewer #2 (Remarks to the Author):

This paper focuses on the structure of a complex between a small HSP found in chloroplasts (Hsp21) and a putative in vivo substrate, 1-deoxy-D-xylulose 5-phosphate synthase (DXPS). The authors isolated a stable complex of Hsp21 bound as a monomer across the dimer interface of DXPS after coexpression of the two proteins in *E. coli* at 37 C. In addition, they present some data aimed at confirming function of Hsp21 in protecting DXPS in vivo and a structure of the Hsp21 chaperone alone as a dodecamer. This represents the first structure of an sHSP with a potential substrate, which is a novel feature of this manuscript. The mode of interaction between the sHSP and substrate are consistent with the existing body of data concerning sHSP-substrate recognition. Specifically, the sHSP binds DXPS that still maintains most structure, The sHSP interacts with substrate regions that are most flexible (especially at high temperature), and it is a dissociated form of the oligomeric sHSP that interacts with substrate. Although dissociated dimers appear to be a major substrate-encounter form for some sHSPs, the observation that a monomer form of the sHSP binds substrate is also supported by observations with other sHSPs. In total, this represents structural confirmation of a large body of work on sHSP-substrate interactions. The novelty lies in having identified specific structural elements showing stable interaction between Hsp21 and DXPS. Unfortunately, as is the case for many sHSP structures, the disposition of the NTR is absent from the structure, and its role unclear, although it contains a major conserved domain of the chloroplast proteins.

While the structural data support many aspects of previous models, there is insufficient data to support the significance or specificity of any in vivo interaction of Hsp21 with DXPS. The in vivo studies are a major shortcoming of the manuscript. The in vivo experiments are incompletely described and controlled. DXPS may be an important substrate of Hsp21, but this needs additional work before it becomes established in the literature. Specific comments on these experiments are detailed below.

- 1) A number of critical controls are missing from the experiments in Figure 1c. These include quantification of punctae of DXPS expressed with native Hsp21 (not fused to GFP), punctae of GFP alone and with DXPS-YFP, and quantification of Hsp21-RFP punctae, both in the presence of native DXPS and DXPS-GFP. The authors also need to be cautious as they are over-expressing these proteins. Do they know the level of the proteins in the chloroplast compared to what is actually present?
- 2) The authors do not explain the rationale for lincomycin treatment, or why this should cause aggregation of DXPS.
- 3) Why haven't the authors tried heat stress instead of lincomycin? Do chloroplasts accumulate HSPs with lincomycin treatment? This is suggested, though not specifically stated in the text when mentioning the effects of lincomycin on gene expression.
- 4) Figure 1d is not very convincing concerning the effect of lincomycin or Hsp21 on the solubility of DXPS-YFP in vivo. The difference in solubility is a few percentage points, with no indication of the replications performed or error in the data.
- 5) The conclusion that the sHSP and DXPS do not colocalize because the sHSP is a "holdase" that is rapidly "handing off" substrate to the other chaperone machinery is a very weak argument.
- 6) Can the authors really claim specificity of the Hsp21-DXPS interaction? They could easily target cytosolic sHSPs to the chloroplast in vivo, and although they would not be found in this compartment in the cell, it would address the specificity and potential differences in the mechanism.
- 7) Fig 1e indicates expression of the sHSP enhances solubility of DXPS at 37 C. This is difficult to conclude from this picture. Do they have any quantitation to support this claim? It would also be easy enough to test other sHSPs for this effect in *E. coli*, again addressing specificity.
- 8) Do they isolate a complex of the sHSP with DXPS when they express the two proteins at 18 C, or only at 37 C. I do not recall that this is addressed in the manuscript.
- 9) It is unclear why the authors did not perform any other assays of Hsp21 protection of DXPS using the purified proteins. Based on Fig. 1e the authors state that Hsp21 would be expected to protect DXPS during heat stress in plants – this is a stretch, and could at least be tested in vitro using many of the assays established for demonstrating sHSP protection of substrates.
- 10) The authors have no data on dissociation of Hsp21 into a monomer, other than finding it in the complex as a monomer.
- 11) The authors state that the sHSP "actively" unfolds itself. I am not sure what "actively" means in this context.

Questions concerning the structural data presentation:

- 1) The authors refer to their isolation of the Hsp21-DXPS complex as "reconstitution". I am not sure that is the right description – it calls to mind taking two purified proteins and putting them back together. Maybe this is OK, but perhaps there is a better term for what they actually did, expressing them together in *E. coli*.
- 2) A significant conserved feature of chloroplast sHSPs is their extended NTR, containing a compared to the plant cytosolic sHSPs. This element is barely mentioned in the paper. In fact, based on the observed interaction with substrate, it would seem that most any sHSP could protect DXPS.
- 3) I found the structural figures less than satisfactory and the figure legends to be minimal. In each case the authors should refer to the structure presented with inclusion of whatever tag is present on the protein. It is hard to know if one is looking at a tagged or untagged version.
- 4) Labeling of N- and C-termini would be helpful, as well as labeling the beta strands in all figures.
- 5) Better figures and discussion of the differences in Hsp21 when bound or not to DXPS would enhance the manuscript. For example, it is unclear why Figure S10 are not in the manuscript

Other issues:

- 1) The authors make the general statement that sHSPs in plants accumulate to up to 1% of total cellular proteins. While this may be true for the total accumulation of sHSPs in the cytosol, relevant to the biology presented here, they should refer to what is known about accumulation of the Hsp21 in the chloroplast.
- 2) Check for misspelling of lincomycin in the text (lincoymycin)

Reviewer #3 (Remarks to the Author):

Small heat shock proteins (sHsps) represent a class of highly conserved molecular chaperones that prevent irreversible unfolding or aggregation of proteins under stress conditions to maintain protein homeostasis. The ability to assemble into oligomers of varying subunit stoichiometries represents a unique feature of sHsps. sHsp oligomers can undergo rapid subunit exchange with the rate of exchange being temperature-dependent, and this structural plasticity is likely essential for the chaperone activity. Activation of sHsp is generally thought to involve the disassembly of inactive oligomers to active sub-oligomeric species through poorly understood mechanisms. Furthermore, sHsps are promiscuous chaperones capable of recognizing a broad spectrum of non-native substrate proteins. Although studied extensively, detailed understanding of substrate recognition and specificity conferred by sHsp remains very limited.

In this manuscript, Yu et al report on the formation of a stable complex between a plastid sHsp, Hsp21, and its natural substrate 1-deoxy-D-xylulose 5 phosphate synthase (DXPS) under heat stress, as well as its cryo-electron microscopy structure at near-atomic resolution. They discovered: (1) monomeric Hsp21 binds across the dimer interface of DXPS and engages in multivalent interactions by recognizing highly dynamic structural elements in DXPS. (2) Hsp21 partly unfolds its central a-crystallin domain to facilitate binding of DXPS, which preserves a native-like structure. Based on these studies, they proposed a mechanism of the anti-aggregation activity of sHsps towards a broad range of substrates. They propose that at normal temperatures, Hsp21 forms dodecamer; in response to elevated temperature, DXPS undergoes thermal fluctuations during which the mobile loop samples multiple conformations rapidly, thereby transiently exposing the chaperone-binding site. Higher temperature increases the rate of dissociation of the Hsp21 dodecamer via the labile interfaces, followed by its partial unfolding concomitant with substantial conformational reorganization to accommodate DXPS. Hsp21 binding thus involves selection of the chaperone-binding site on DXPS from ensembles of thermally accessible states as well as structural changes in Hsp21 triggered by multivalent engagement to DXPS.

Major concerns:

1. Although co-expression experiment demonstrated that the expression of Hsp21 increased the solubility of DXPS in vivo (Fig.1), it is insufficient to conclude that the increase of solubility of DXPS is due to a direct interaction between Hsp21 and DXPS. It is necessary to demonstrate that Hsp21 can prevent the aggregation of DXPS by direct interaction in vitro using purified Hsp21 and DXPS.
2. The discovery that monomeric Hsp21 forms a complex with substrate DXPS is interesting, which is contradictory to the hypothesis that dimer acts as the functional unit of sHsps. The authors need more experiments to verify the mechanism they proposed. One experiment is that the authors should purify Hsp21 in the form of dodecamer and DXPS as dimer, mix them together at elevated temperatures (37 °C, for example), then observe whether the dodecamer dissociate into monomers and form the monomeric Hsp21-DXPS dimer complex.

Structural basis of substrate recognition and thermal protection by a small heat shock protein

Authors' response

We thank the editor and the three reviewers for taking the time to critically review our manuscript and provide excellent comments. We have conducted additional experiments and made a number of changes in response to these comments, which significantly improved the manuscript. Please find our point-by-point response below.

Reviewer #1:

This manuscript describes arguably the first structural analysis of a substrate:small heat-shock protein complex, revealing considerable insight into the mechanism by which these proteins work. This is an important piece of work that should be published in Nat Comms.

My main criticism is the the presentation of the structural figures - in general I found it quite hard to "read" them well - seeing the differences the authors allude to (e.g. unstructured B2 strand upon binding, the difference in dimerisation between HSP21 and HSP16.9 etc) is difficult because labelling is sparse, alignments are confusing (and sometimes seem poor). In general, I would urge the authors to look at the structural figures carefully to make sure that everything they wish to convey can be clearly seen, and whether they have labelled strands and relevant residues clearly.

RESPONSE: We thank the reviewer for his/her overall positive recommendation. We have improved the clarity of the figures, which include Fig. 2-4 and Supplementary Figure 9 (updated figure numbers).

Minor comments:

1) The authors say DXPS is important for antimalarials etc - presumably not in arabidopsis – perhaps just make clear this is referring to the human homologue

RESPONSE: We have now clarified this in the text (line 92-94).

2) The difference in soluble v pellet change upon adding HSP21 (Fig., 1d) is very subtle. Are the authors sure that the increase from 87% to 92% are real?

RESPONSE: To ensure the change in the difference of the soluble and insoluble fractions is real and reproducible, we performed a total of three independent replicates of the experiments and conducted a test of statistical significance to support our conclusion (Supplementary Figure 1d-f).

3) Fig 3 would be helped by a comparison of the dimers of HSP21 and HSP16.9.

RESPONSE: We updated Fig. 3 by including a comparison of the dimeric and non-dimeric interface of Hsp21 and wheat Hsp16.9 and discuss the similarity and differences of the binding modes (line 245-258).

- 4) The discussion about the B2 strand being disordered upon binding is interesting in light of the findings of Collier et al, Sci Adv, 2019 (eaav8421) which noted phosphorylation dependent disorder in the B2 strand of HSPB1

RESPONSE: Thanks for the comment and we fully agree with the Reviewer that this is a very interesting observation emerged from our structural study. We have also included the suggested the citation in the text (line 289-291).

Reviewer #2:

This paper focuses on the structure of a complex between a small HSP found in chloroplasts (Hsp21) and a putative in vivo substrate, 1-deoxy-D-xylulose 5-phosphate synthase (DXPS). The authors isolated a stable complex of Hsp21 bound as a monomer across the dimer interface of DXPS after coexpression of the two proteins in *E. coli* at 37 C. In addition, they present some data aimed at confirming function of Hsp21 in protecting DXPS in vivo and a structure of the Hsp21 chaperone alone as a dodecamer. This represents the first structure of an sHSP with a potential substrate, which is a novel feature of this manuscript. The mode of interaction between the sHSP and substrate are consistent with the existing body of data concerning sHSP-substrate recognition. Specifically, the sHSP binds DXPS that still maintains most structure, The sHSP interacts with substrate regions that are most flexible (especially at high temperature), and it is a dissociated form of the oligomeric sHSP that interacts with substrate. Although dissociated dimers appear to be a major substrate-encounter form for some sHSPs, the observation that a monomer form of the sHSP binds substrate is also supported by observations with other sHSPs. In total, this represents structural confirmation of a large body of work on sHSP-substrate interactions. The novelty lies in having identified specific structural elements showing stable interaction between Hsp21 and DXPS. Unfortunately, as is the case for many sHSP structures, the disposition of the NTR is absent from the structure, and its role unclear, although it contains a major conserved domain of the chloroplast proteins.

While the structural data support many aspects of previous models, there is insufficient data to support the significance or specificity of any in vivo interaction of Hsp21 with DXPS. The in vivo studies are a major shortcoming of the manuscript. The in vivo experiments are incompletely described and controlled. DXPS may be an important substrate of Hsp21, but this needs additional work before it becomes established in the literature. Specific comments on these experiments are detailed below.

RESPONSE: We appreciate the Reviewer's overall positive attitude towards our work. We fully agree with the Reviewer's that our study represents the first structure of a sHsp-substrate complex that reveals many novel features and new biological insights. With regards to the in vivo studies, we have now included additional experimental data to support the interaction of Hsp21 and DXPS, as described below.

- 1) A number of critical controls are missing from the experiments in Figure 1c. These include quantification of punctae of DXPS expressed with native Hsp21 (not fused to GFP), punctae of GFP alone and with DXPS-YFP, and quantification of Hsp21-RFP punctae, both in the presence of native DXPS and DXPS-GFP. The authors also need to be cautious as they are over-expressing these proteins. Do they know the level of the proteins in the chloroplast compared to what is actually present?

RESPONSE: The original Fig. 1 has been updated and moved to the Supplementary Section as Supplementary Figure 1. All appropriate controls of the confocal experiments and quantification were now added in Supplementary Figure 1a-c, which include the fusion tags alone fused to TP (transit peptide - to facilitate the import of the tags into the chloroplast) (row 1 and 2, left and right panels in a) and Hsp21-His and DXPS-His as native proteins (row 4-5, left and right panels in a). We are aware that we were overexpressing the proteins and we have stated in manuscript (line 99-104). In fact, by taking advantage of the unique aggregation property of DXPS in response to overexpression, we were able to assay the aggregation suppression activity of Hsp21 against DXPS in vivo, which was further confirmed by biochemical and structural studies. As we do not have the antibody against either the chaperone or the substrate, we do not know the absolute level of the overexpressed proteins in the chloroplast.

2) The authors do not explain the rationale for lincomycin treatment, or why this should cause aggregation of DXPS.

RESPONSE: Lincomycin treatment causes DXPS aggregation as a result of disruption and consequently reduction of the activity of the Clp proteolytic complex (ref 28). The explanation for the use of the lincomycin treatment to trigger the aggregation of DXPS can be found in line 84-89 and 94-99.

3) Why haven't the authors tried heat stress instead of lincomycin? Do chloroplasts accumulate HSPs with lincomycin treatment? This is suggested, though not specifically stated in the text when mentioning the effects of lincomycin on gene expression.

RESPONSE: As per the Reviewer's suggestions, we have additionally performed the confocal imaging and fractionation analysis under heat stress conditions (Fig. 1a-f) with all the appropriate controls and quantification as in Supplementary Figure 1. In order to be able to directly observe the changes in the level of DXPS aggregates in response to elevated temperature, the heat stress experiments were performed together with lincomycin treatment because inhibition of chloroplast translation/PGE disrupts the Clp complex, thereby preventing the degradation of DXPS aggregates. Lincomycin treatment also promotes the accumulation of Hsp21 (ref. 28) and is stated in the text (line 84-87).

4) Figure 1d is not very convincing concerning the effect of lincomycin or Hsp21 on the solubility of DXPS-YFP in vivo. The difference in solubility is a few percentage points, with no indication of the replications performed or error in the data.

RESPONSE: Replications and quantification analysis are now included to confirm our previous observation that Hsp21 suppresses the aggregation of DXPS in vivo, for both with (Fig. 1a-f) and without heat treatment (Supplementary Figure 1a-f).

5) The conclusion that the sHSP and DXPS do not colocalize because the sHSP is a "holdase" that is rapidly "handing off" substrate to the other chaperone machinery is a very weak argument.

RESPONSE: Our new fractionation analysis under heat stress conditions revealed that although the presence of Hsp21 rendered DXPS soluble, the soluble DXPS did not co-fractionate with Hsp21 in the supernatant, suggesting that Hsp21 hands off DXPS to other chaperone machinery for its active dissolution and/or refolding in vivo (line 123-130).

- 6) Can the authors really claim specificity of the Hsp21-DXPS interaction? They could easily target cytosolic sHSPs to the chloroplast in vivo, and although they would not be found in this compartment in the cell, it would address the specificity and potential differences in the mechanism.

RESPONSE: We thank the Reviewer for the suggested experiment. In the original manuscript, we have used the term 'specificity' only once throughout, which is in the Introduction section ("Although studied extensively, detailed understanding of substrate recognition and specificity conferred by sHsp remains very limited"). In this context, we used the term 'specificity' to describe the unique structural binding mode of sHsp to its substrates, which does not imply that DXPS can only interact with Hsp21 but not other sHsps, or vice versa. We apologize for the confusion caused and we have now removed this term in the Introduction. Our current study provides a structural snapshot of Hsp21 interacting with DXPS in a unique binding mode by use of cryo-EM, but whether or not DXPS can interact with another sHsp is beyond the scope of this study.

- 7) Fig 1e indicates expression of the sHSP enhances solubility of DXPS at 37 C. This is difficult to conclude from this picture. Do they have any quantitation to support this claim? It would also be easy enough to test other sHSPs for this effect in E. coli, again addressing specificity.

RESPONSE: We have now included quantification to support this claim with replications (Fig. 1g, h). As discussed above, we made no attempt to address the specificity of the Hsp21-DXPS interaction in this manuscript.

- 8) Do they isolate a complex of the sHSP with DXPS when they express the two proteins at 18 C, or only at 37 C. I do not recall that this is addressed in the manuscript.

RESPONSE: We isolated the complex from E. coli grown at 37 °C, a condition that promotes thermally induced DXPS aggregation and at the same time, soluble Hsp21-DXPS complex formation (line 142-145).

- 9) It is unclear why the authors did not perform any other assays of Hsp21 protection of DXPS using the purified proteins. Based on Fig. 1e the authors state that Hsp21 would be expected to protect DXPS during heat stress in plants – this is a stretch, and could at least be tested in vitro using many of the assays established for demonstrating sHSP protection of substrates.

RESPONSE: We thank the Reviewer for the suggested experiment. In this revised manuscript, we have performed the aggregation suppression assay using purified Hsp21 and DXPS proteins (Fig. 1i, j) and our in vivo and in vitro experiments all arrived at a same conclusion that Hsp21 can suppress the aggregation of DXPS.

- 10) The authors have no data on dissociation of Hsp21 into a monomer, other than finding it in the complex as a monomer.

RESPONSE: We developed the crosslinking assay to directly probe the dodecamer-monomer dissociation of Hsp21 and observed an increase of Hsp21 monomers at elevated temperature (Supplementary Figure 11).

11) The authors state that the sHSP “actively” unfolds itself. I am not sure what “actively” means in this context.

RESPONSE: We have removed the term ‘actively’ in the Discussion section (line 320-322).

Questions concerning the structural data presentation:

1) The authors refer to their isolation of the Hsp21-DXPS complex as “reconstitution”. I am not sure that is the right description – it calls to mind taking two purified proteins and putting them back together. Maybe this is OK, but perhaps there is a better term for what they actually did, expressing them together in *E. coli*.

RESPONSE: We thank the Reviewer for the suggestion and therefore we have replaced the original sentence with the more correct description regarding the method of isolation of the complex (line 142-145).

2) A significant conserved feature of chloroplast sHSPs is their extended NTR, containing a compared to the plant cytosolic sHSPs. This element is barely mentioned in the paper. In fact, based on the observed interaction with substrate, it would seem that most any sHSP could protect DXPS.

RESPONSE: We have added further description about the NTRs of chloroplast sHsps compared to that of the cytosolic sHsps in plants (line 215-221).

3) I found the structural figures less than satisfactory and the figure legends to be minimal. In each case the authors should refer to the structure presented with inclusion of whatever tag is present on the protein. It is hard to know if one is looking at a tagged or untagged version.

RESPONSE: We have updated the figures and expanded the corresponding legends where necessary.

4) Labeling of N- and C-termini would be helpful, as well as labeling the beta strands in all figures.

RESPONSE: The labels of the beta strands and C-terminus of Hsp21 are now included in all structure figures. The NTRs are almost completely disordered in both the Hsp21-DXPS and Hsp21 dodecamer structures and therefore the N-terminus is not labeled.

5) Better figures and discussion of the differences in Hsp21 when bound or not to DXPS would enhance the manuscript. For example, it is unclear why Figure S10 are not in the manuscript

RESPONSE: An improved version of the figure (now Fig. 4, replacing the original Supplementary Figure 10) and a more detailed discussion of the differences (line 284-298) in Hsp21 when bound to DXPS or not are now included.

Other issues:

1) The authors make the general statement that sHSPs in plants accumulate to up to 1% of total cellular proteins. While this may be true for the total accumulation of sHSPs in the cytosol, relevant to the biology presented here, they should refer to what is known about accumulation of the Hsp21 in the chloroplast.

RESPONSE: In a previous study, the Hsp21 level has been estimated to reach ~0.05% of total soluble proteins in the chloroplast in heat-stressed leaves. We have included this information with the corresponding source of reference (line 115-116).

2) Check for misspelling of lincomycin in the text (lincoymycin)

RESPONSE: The typo has been corrected in the manuscript.

Reviewer #3:

Small heat shock proteins (sHsps) represent a class of highly conserved molecular chaperones that prevent irreversible unfolding or aggregation of proteins under stress conditions to maintain protein homeostasis. The ability to assemble into oligomers of varying subunit stoichiometries represents a unique feature of sHsps. sHsp oligomers can undergo rapid subunit exchange with the rate of exchange being temperature-dependent, and this structural plasticity is likely essential for the chaperone activity. Activation of sHsp is generally thought to involve the disassembly of inactive oligomers to active sub-oligomeric species through poorly understood mechanisms. Furthermore, sHsps are promiscuous chaperones capable of recognizing a broad spectrum of non-native substrate proteins. Although studied extensively, detailed understanding of substrate recognition and specificity conferred by sHsp remains very limited.

In this manuscript, Yu et al report on the formation of a stable complex between a plastid sHsp, Hsp21, and its natural substrate 1-deoxy-D-xylulose 5 phosphate synthase (DXPS) under heat stress, as well as its cryo-electron microscopy structure at near-atomic resolution. They discovered: (1) monomeric Hsp21 binds across the dimer interface of DXPS and engages in multivalent interactions by recognizing highly dynamic structural elements in DXPS. (2) Hsp21 partly unfolds its central α -crystallin domain to facilitate binding of DXPS, which preserves a native-like structure. Based on these studies, they proposed a mechanism of the anti-aggregation activity of sHsps towards a broad range of substrates. They propose that at normal temperatures, Hsp21 forms dodecamer; in response to elevated temperature, DXPS undergoes thermal fluctuations during which the mobile loop samples multiple conformations rapidly, thereby transiently exposing the chaperone-binding site. Higher temperature increases the rate of dissociation of the Hsp21 dodecamer via the labile interfaces, followed by its partial unfolding concomitant with substantial conformational reorganization to accommodate DXPS. Hsp21 binding thus involves selection of the chaperone-binding site on DXPS from ensembles of thermally accessible states as well as structural changes in Hsp21 triggered by multivalent engagement to DXPS.

Major concerns:

1) Although co-expression experiment demonstrated that the expression of Hsp21 increased the solubility of DXPS in vivo (Fig.1), it is insufficient to conclude that the increase of solubility of DXPS is due to a direct interaction between Hsp21 and DXPS. It is necessary

to demonstrate that Hsp21 can prevent the aggregation of DXPS by direct interaction in vitro using purified Hsp21 and DXPS.

RESPONSE: We thank the Reviewer for the suggested experiment. Using purified proteins and in vitro aggregation suppression and crosslinking assays, we have directly demonstrated the direct interaction between Hsp21 and DXPS (Fig. 1i, j and Supplementary Figure 11b).

2) The discovery that monomeric Hsp21 forms a complex with substrate DXPS is interesting, which is contradictory to the hypothesis that dimer acts as the functional unit of sHsps. The authors need more experiments to verify the mechanism they proposed. One experiment is that the authors should purify Hsp21 in the form of dodecamer and DXPS as dimer, mix them together at elevated temperatures (37 °C, for example), then observe whether the dodecamer dissociate into monomers and form the monomeric Hsp21-DXPS dimer complex.

RESPONSE: We thank for the Reviewer for the thoughtful experiment. We have performed this suggested experiment using purified Hsp21 and DXPS, and then probed the dodecamer-monomer dissociation of Hsp21 and complex formation of Hsp21-DXPS by use of a crosslinking assay (Supplementary Figure 11). These data fully support our cryo-EM analysis.

REVIEWERS' COMMENTS

Reviewer #1 (Remarks to the Author):

All my comments have been addressed. In my opinion the work is ready for acceptance at Nature Comms.

Reviewer #2 (Remarks to the Author):

This paper describes, by cryo EM, the structural interaction of an sHSP monomer from the chloroplast-localized Hsp21 with an endogenous substrate, the enzyme DXPS, involved in the MEP pathway of carotenoid biosynthesis. It also includes a cryo-EM description of the sHSP free dodecamer. Results are interesting and pose new questions about the specific role of Hsp21 in chloroplasts as well as different ways in which different sHSPs may act to interact with substrates. There are a number of points listed below that need to be addressed, and overall, it is important that the authors clarify better the limitations of their data (see some specifics below), and also that this is only one example of how sHSP-substrate complexes may be formed. This is important as the bulk of the interactions in their system are not the complex shown, but rather material that is insoluble in their assays.

Fig. 1. The authors state that the punctate Hsp21 appearance is due to nucleoid association. However, just because the Hsp21-RFP is in specific regions of the chloroplast does not confirm it is associated with nucleoids. The DXPS is also in discrete foci – which they call aggregates. They need to delete the statement that this punctate localization corresponds to association with nucleoids without colocalization of a nucleoid marker.

Fig. 1b and c. The authors provide the number of puncta quantified, but do not state the number of independent experiments from which the data were derived. It is important that more than one, and preferably more than three were involved.

Fig. 1e, h, j. Please add actual data points to the bar graphs.

Fig. 1g. Very difficult to see how data as seen on this gel could be accurately quantified. Signals all appear oversaturated.

Fig. 1i. For clarity state that Hsp21:DXPS molar ratio is monomer/monomer, if that is the case, rather than requiring that assumption.

Fig. S1e and f. Please add actual data points to the bar graph in e and f.

Fig. 1. Although this information is undoubtedly in the methods, the authors should state the length of the 37C treatment in the figure legend.

Although perhaps not required within the scope of this paper, it would be of significant interest to show specificity of this interaction to Hsp21 by performing the experiment with any other plant or other sHSP. While extensive for the in vivo analyses, it would be relatively trivial for an in vivo study in *E. coli* or for an in vitro study as in Fig. 1g and 1i, respectively.

Line 126. The authors state: "The fact that the soluble DXPS was not found in complex with Hsp21 suggests a functional cooperation of Hsp21 with ATP-dependent chaperones for the active dissolution and/or refolding of DXPS in the cell, which likely explains the apparent lack of colocalization between Hsp21 and DXPS under confocal microscopy". This is an over-reaching explanation. There is also inconsistency in what the authors are referring to as "soluble" vs "aggregated". From the microscopy in plant cells, DXPS is always in "punctae". So are these insoluble aggregates? Or are they both soluble and insoluble? Or is there soluble protein that is not visible because it is diluted out compared to the punctae?

What is difference between the "in cell" observations and the in vitro observations? That is, in the plant cell or *E. coli*, at 37 C most of the DXPS remains insoluble, while in vitro, almost complete solubility is observed?

Can the authors clarify that the material they are imaging is equivalent to the ~5% soluble material seen in Fig. 1g in the second from last lane? I think that this is a relevant detail for others interested

in what types of complexes are formed by these sHSPs and their mechanism of action.

Supplementary Fig. 6. Can the authors supply some kind of quantitative data on the RMS differences?

Supplementary Fig. 11. Not a very convincing assay of dissociation. Not sure this adds to the authors argument.

Supplementary Fig. 12. I am not sure what the authors are trying to demonstrate with these data. The negative staining in a and b are vastly different. The authors need to be more clear as to what structural perturbation in the figure is actually consistent with the dissociation. There are other ways to assess dissociation.

The authors conclude: "Insights obtained from this study provide opportunities for the potential modulation of the MEP pathway for broad industrial and pharmaceutical applications through targeting the rate-limiting enzyme and its chaperone." I find this statement extremely vague. Either be more specific or delete.

Similar to the above: "It also sheds light on the additional role of ATP-dependent chaperones for the subsequent release and reactivation of sHsp from the sHsp-substrate complex." I do not know what data in this paper they are referring to in drawing this conclusion.

Other:

It is an overstatement of what is known to say that sHSPs are a "first line of defense" against protein aggregation, especially in the case of Hsp21, which is not produced until heat stress has occurred – so it is not there as "the first line". I think the first statement in the abstract is therefore hyperbola. In a number of places the manuscript would benefit from editing for English language usage. Is it really necessary to use the abbreviation PGE? This seems unnecessary and distracting.

Reviewer #3 (Remarks to the Author):

The authors have performed experiments I suggested and fully addressed the issues I raised. The manuscript has reached the criteria for publication in Nature Communications.

Structural basis of substrate recognition and thermal protection by a small heat shock protein

Authors' response

We thank the editor and the three reviewers for their support and excellent comments. We have revised the manuscript according to the Reviewer #2's comments. Please find our point-by-point response below.

Reviewer #2:

This paper describes, by cryo EM, the structural interaction of an sHSP monomer from the chloroplast-localized Hsp21 with an endogenous substrate, the enzyme DXPS, involved in the MEP pathway of carotenoid biosynthesis. It also includes a cryo-EM description of the sHSP free dodecamer. Results are interesting and pose new questions about the specific role of Hsp21 in chloroplasts as well as different ways in which different sHSPs may act to interact with substrates. There are a number of points listed below that need to be addressed, and overall, it is important that the authors clarify better the limitations of their data (see some specifics below), and also that this is only one example of how sHSP-substrate complexes may be formed. This is important as the bulk of the interactions in their system are not the complex shown, but rather material that is insoluble in their assays.

RESPONSE: We thank the Reviewer for his/her positive comment and the constructive comments.

1) Fig. 1. The authors state that the punctate Hsp21 appearance is due to nucleoid association. However, just because the Hsp21-RFP is in specific regions of the chloroplast does not confirm it is associated with nucleoids. The DXPS is also in discrete foci – which they call aggregates. They need to delete the statement that this punctate localization corresponds to association with nucleoids without colocalization of a nucleoid marker.

RESPONSE: We agree with the Reviewer. Since we did not perform colocalization analysis of Hsp21 with a nucleoid marker in current study but simply cited the previously published report, therefore we have modified the statements in the text accordingly (line 100-104).

2) Fig. 1b and c. The authors provide the number of puncta quantified, but do not state the number of independent experiments from which the data were derived. It is important that more than one, and preferably more than three were involved.

RESPONSE: At least two independent experiments from which the data were derived from, and this information has been added to the Fig. 1 legend.

3) Fig. 1e, h, j. Please add actual data points to the bar graphs.

RESPONSE: The actual data points have now been added to the bar graphs.

4) Fig. 1g. Very difficult to see how data as seen on this gel could be accurately quantified. Signals all appear oversaturated.

RESPONSE: The signal quantification was performed using the ImageJ software. According to the software, the band intensities were not oversaturated.

5) Fig. 1i. For clarity state that Hsp21:DXPS molar ratio is monomer/monomer, if that is the case, rather than requiring that assumption.

RESPONSE: Yes indeed, the molar ratio is monomer/monomer. We have added this information to the respective figure legend.

6) Fig. S1e and f. Please add actual data points to the bar graph in e and f.

RESPONSE: The actual data points have now been added to the bar graphs.

7) Fig. 1. Although this information is undoubtedly in the methods, the authors should state the length of the 37C treatment in the figure legend.

RESPONSE: The length of all the temperature treatments are already included in the Fig. 1 legend.

8) Although perhaps not required within the scope of this paper, it would be of significant interest to show specificity of this interaction to Hsp21 by performing the experiment with any other plant or other sHSP. While extensive for the in vivo analyses, it would be relatively trivial for an in vivo study in *E. coli* or for an in vitro study as in Fig. 1g and 1i, respectively.

RESPONSE: We thank the Reviewer for the suggested experiments.

9) Line 126. The authors state: “The fact that the soluble DXPS was not found in complex with Hsp21 suggests a functional cooperation of Hsp21 with ATP-dependent chaperones for the active dissolution and/or refolding of DXPS in the cell, which likely explains the apparent lack of colocalization between Hsp21 and DXPS under confocal microscopy”. This is an over-reaching explanation. There is also inconsistency in what the authors are referring to as “soluble” vs “aggregated”. From the microscopy in plant cells, DXPS is always in “punctae”. So are these insoluble aggregates? Or are they both soluble and insoluble? Or is there soluble protein that is not visible because it is diluted out compared to the punctae?

RESPONSE: According to the Reviewer’s suggestion, we have modified the text to provide the plausible explanations to our observation that the Hsp21-DXPS interaction was not detected in the soluble fraction by fractionation followed by Western analysis (line 130-133). As also suggested by the Reviewer, the apparent lack of colocalization between Hsp21 and DXPS under confocal microscopy was likely due to fluorescence signals of the Hsp21-DXPS complexes being diluted out compared to that of the punctate spots; we have now included this statement in the text (line 109-112). Throughout the text, we have consistently used the terms soluble and insoluble fractions to represent soluble complexes and (insoluble) protein aggregates, respectively. Since both lincomycin and heat treatment led to an increase in the total DXPS punctate spot area as well as the amount of materials in the insoluble fractions, we believe that the DXPS punctae represent insoluble aggregates, which is also consistent with the published reports (29, 32-33).

- 10) What is difference between the “in cell” observations and the in vitro observations? That is, in the plant cell or E. coli, at 37 C most of the DXPS remains insoluble, while in vitro, almost complete solubility is observed?

RESPONSE: This difference is now described in the text “At the molar ratio of Hsp21 monomer to DXPS monomer of 12:1, near complete solubilization of DXPS was observed, in contrast to the ~10-20% soluble DXPS observed in vivo in the presence of Hsp21 (Fig. 1e, h and j).” (line 140-143).

- 11) Can the authors clarify that the material they are imaging is equivalent to the ~5% soluble material seen in Fig. 1g in the second from last lane? I think that this is a relevant detail for others interested in what types of complexes are formed by these sHSPs and their mechanism of action.

RESPONSE: The soluble materials isolated for cryo-EM imaging is equivalent to ~20% of the total expressed DXPS proteins. This information has now been added in the text (line 151-153).

- 12) Supplementary Fig. 6. Can the authors supply some kind of quantitative data on the RMS differences?

RESPONSE: We have now provided the C(alpha) RMSD in the Supplementary Figure 6.

- 13) Supplementary Fig. 11. Not a very convincing assay of dissociation. Not sure this adds to the authors argument.

RESPONSE: While there are a number of ways to demonstrate dissociation of sHsp oligomers, we have chosen the crosslinking assay according to the published study on Hsp21 with a model substrate (Ref. 19). Our result showed that the amount of Hsp21 monomers was increased at higher temperature compared to the dodecameric form, and that the amount of Hsp21 monomers was decreased in the presence of its substrate DXPS along with the appearance of the Hsp21-DXPS complex, the identity of which was confirmed by mass spectrometry.

- 14) Supplementary Fig. 12. I am not sure what the authors are trying to demonstrate with these data. The negative staining in a and b are vastly different. The authors need to be more clear as to what structural perturbation in the figure is actually consistent with the dissociation. There are other ways to assess dissociation.

RESPONSE: To explain this clearly, we have modified the text to state that “The structural perturbation of the Hsp21 dodecamer leading to an overall size reduction of the oligomer induced by elevated temperature was also directly visualized by negative stain electron microscopy (EM) (Supplementary Figure 12), providing further consistent evidence for Hsp21 dodecamer dissociation at high temperature.” (Line 286-289).

- 15) The authors conclude: “Insights obtained from this study provide opportunities for the potential modulation of the MEP pathway for broad industrial and pharmaceutical applications through targeting the rate-limiting enzyme and its chaperone.” I find this statement extremely vague. Either be more specific or delete.

RESPONSE: We have modified the statement accordingly (line 326-328).

- 16) Similar to the above: “It also sheds light on the additional role of ATP-dependent chaperones for the subsequent release and reactivation of sHsp from the sHsp-substrate complex.” I do not know what data in this paper they are referring to in drawing this conclusion.

RESPONSE: We have deleted this statement.

- 17) Other: It is an overstatement of what is known to say that sHSPs are a “first line of defense” against protein aggregation, especially in the case of Hsp21, which is not produced until heat stress has occurred – so it is not there as “the first line”. I think the first statement in the abstract is therefore hyperbola.

RESPONSE: We have deleted this statement and modified the abstract.

- 18) In a number of places the manuscript would benefit from editing for English language usage. Is it really necessary to use the abbreviation PGE? This seems unnecessary and distracting.

RESPONSE: We have abandoned the use of the abbreviation PGE in the manuscript.